# Tissue Fraction Correction and Visual Analysis Increase Diagnostic Sensitivity in Predicting Malignancy of Ground-Glass Nodules on [^18^F]FDG PET/CT: A Bicenter Retrospective Study

**DOI:** 10.3390/diagnostics12051292

**Published:** 2022-05-23

**Authors:** Yun Hye Song, Jung Won Moon, Yoo Na Kim, Ji Young Woo, Hye Joo Son, Suk Hyun Lee, Hee Sung Hwang

**Affiliations:** 1Department of Radiology, Hallym University Kangnam Sacred Heart Hospital, College of Medicine, Hallym University, Seoul 07441, Korea; yhsong31@naver.com (Y.H.S.); moonyard@hallym.or.kr (J.W.M.); ynkim13@hallym.or.kr (Y.N.K.); baccas@hallym.or.kr (J.Y.W.); 2Department of Nuclear Medicine, Dankook University Medical Center, Cheonan 31116, Korea; neuroscience@dankook.ac.kr; 3Department of Nuclear Medicine, Hallym University Sacred Heart Hospital, College of Medicine, Hallym University, Anyang 14068, Korea

**Keywords:** ground-glass nodule, tissue fraction correction, visual analysis, [^18^F]FDG positron emission tomography/computed tomography

## Abstract

We investigated the role of [^18^F]FDG positron emission tomography/computed tomography (PET/CT) in evaluating ground-glass nodules (GGNs) by visual analysis and tissue fraction correction. A total of 40 pathologically confirmed ≥1 cm GGNs were evaluated visually and semiquantitatively. [^18^F]FDG uptake of GGN distinct from background lung activity was considered positive in visual analysis. In semiquantitative analysis, we performed tissue fraction correction for the maximum standardized uptake value (SUV_max_) of GGN. Of the 40 GGNs, 25 (63%) were adenocarcinomas, 9 (23%) were minimally invasive adenocarcinomas (MIAs), and 6 (15%) were adenocarcinomas in situ (AIS). On visual analysis, adenocarcinoma showed the highest positivity rate among the three pathological groups (88%, 44%, and 17%, respectively). Both SUV_max_ and tissue-fraction–corrected SUV_max_ (SUV_maxTF_) were in the order of adenocarcinoma > MIA > AIS (*p* = 0.033 and 0.018, respectively). SUV_maxTF_ was significantly higher than SUV_max_ before correction (2.4 [1.9–3.0] vs. 1.3 [0.8–1.8], *p* < 0.001). When using a cutoff value of 2.5, the positivity rate of GGNs was significantly higher in SUV_maxTF_ than in SUV_max_ (50% vs. 5%, *p* < 0.001). The diagnostic sensitivity of [^18^F]FDG PET/CT in predicting the malignancy of lung GGN was improved by tissue fraction correction and visual analysis.

## 1. Introduction

With the development of thin section and high-resolution chest computed tomography (CT) [1,2,3], the detection rate of ground-glass nodules (GGN) has also increased. Lesions of various etiologies can be seen as GGNs, representatively benign lesions—such as inflammatory diseases, focal hemorrhages, and fibroses—and precancerous lesions such as atypical adenomatous hyperplasia. In addition, malignancies, such as adenocarcinoma in situ (AIS), minimally invasive adenocarcinoma (MIA), and some invasive adenocarcinomas have been reported as GGNs [4,5,6].

2-Deoxy-2-[^18^F]fluoro-D-glucose ([^18^F]FDG) positron emission tomography/computed tomography (PET/CT) is known to exhibit high sensitivity and specificity in differentiating benign from malignant solid lung nodules [7,8,9]. Therefore, [^18^F]FDG PET/CT is strongly recommended as a method to evaluate a single solid lung nodule according to the Fleischner Society 2017 Guidelines [10]. However, the role of [^18^F]FDG PET/CT in evaluating a subsolid nodule remains unclear [11,12,13].

Several studies have shown that non-small-cell lung cancer (NSCLC) expressed as subsolid nodules has lower [^18^F]FDG uptake than other types of NSCLC. In particular, the false-negative rate of a malignant pure GGN has been reported as high as 90–100% [13,14,15,16]. One of the reasons for the low [^18^F]FDG uptake of malignant subsolid nodules is that [^18^F]FDG is not distributed in the air portion within the nodule, which may underestimate the [^18^F]FDG uptake of the solid portion.

Lambrou et al. presented a method to correct the air fraction of the lung by measuring the Hounsfield units (HUs) in interstitial lung disease. The air fraction may be heterogeneous depending on the severity of interstitial lung disease; therefore, the intention was to correct this effect on [^18^F]FDG uptake [17]. Application of this method to pure GGNs might enable the measurement of the [^18^F]FDG uptake of the solid portion of the nodule, excluding the air fraction.

This study aimed to investigate the role of [^18^F]FDG PET/CT in evaluating GGNs and determine if tissue fraction correction is beneficial for interpreting [^18^F]FDG uptake.

## 2. Materials and Methods

### 2.1. Subjects

This study was approved by the Institutional Review Boards (IRB) of Kangnam Sacred Heart Hospital (IRB no. 2021-05-026) and Hallym Sacred Heart Hospital (IRB no. 2021-08-032). The IRBs waived the requirement of patient informed consent owing to the retrospective nature of this study. Among the patients who underwent chest CT at Kangnam Sacred Heart Hospital from June 2012 to December 2020 and Hallym University Sacred Heart Hospital from November 2013 to December 2020 and exhibited pure GGNs of ≥1 cm, we analyzed those who underwent [^18^F]FDG PET/CT within 90 days of the chest CT (Figure 1). The patients’ age at diagnosis, sex, smoking history, date of chest CT, date of PET/CT, date and method of pathological confirmation, and final pathology were obtained from electronic medical records. The size and location of GGNs were obtained through chest CT data. All GGNs were pathologically confirmed. No patients with benign or malignant tumor could be determined by imaging follow-up for >5 years.

### 2.2. PET/CT Imaging Protocol

We acquired [^18^F]FDG PET/CT images under the following conditions. Before PET/CT, the patient fasted for >6 h and was injected with 5.18 MBq/kg (0.14 mCi/kg) of [^18^F]FDG. The blood glucose level was controlled to be <8.33 mmol/L (150 mg/dL). PET/CT images were acquired approximately 60 min after [^18^F]FDG injection using a Gemini TF 16 PET/CT scanner (Philips Healthcare, Cambridge, MA, USA) and Gemini TF 64 PET/CT scanner (Philips Healthcare, Cambridge, MA, USA). After the initial low-dose CT (120 kVp, 50 mAs, 4 mm slice thickness) scan, PET images were acquired in 3D mode from the skull base to mid-thigh at 7–10 beds, 2 min each. The PET images were reconstructed using the 3D row-action maximum likelihood algorithm and the iterative ordered subsets expectation maximization algorithm (three iterations, 33 subsets, and no filtering), and CT-based attenuation correction was performed. Kangnam Sacred Heart Hospital and Hallym University Sacred Heart Hospital used PET/CT scanners with the same PET resolution and followed the same PET/CT imaging protocol.

### 2.3. PET/CT Image Analysis

Two experienced nuclear medicine board-certified physicians (S.H.L and H.J.S) performed visual analysis. The GGN was considered positive if there was [^18^F]FDG uptake distinct from background lung activity. If the results were discordant, the two physicians reviewed them together to reach a consensus.

For semiquantitative analysis, the maximum standardized uptake value (SUV_max_) was measured on a workstation (Advantage Workstation 4.7, GE Healthcare, Chicago, IL, USA) by placing a volume of interest over each GGN. For the tissue fraction correction of SUV_max_, the following assumptions were made:By adopting the method of Lambrou et al., the SUV of the solid portion within the GGN can be obtained by excluding the air fraction in which [^18^F]FDG is not distributed.This study was conducted on pure GGNs only, and we assumed that the density within a GGN was homogeneous.

The formula for SUV_max_ correction by Lambrou et al. is as follows:The tissue fraction of the GGN is k, and the air fraction is (1 − k).The HU of a GGN (HU_GGN_) can be expressed as follows:
HU_GGN_ = k HU_Tissue_ + (1 − k) HU_Air_Converting to the expression for k, we find the following:
k = (HU_GGN_ − HU_Air_)/(HU_Tissue_ − HU_Air_)We assumed that the HU of the lung tissue fraction of the GGN would be similar to that of other solid organs, such as the liver; therefore, we assigned a value of 50 to HUTissue. HUAir was −1000. The HUGGN of each GGN was measured on low-dose precontrast CT images of PET/CT because many patients only had enhanced chest CT images.If SUV_max_ is divided by k, the tissue-fraction–corrected SUV_max_ (SUV_maxTF_) excluding the air fraction can be obtained.

SUV_maxTF_ = SUV_max_/k

We set the cutoff value of SUV_max_ and SUV_maxTF_ to 2.5, which is commonly used [18].

### 2.4. Statistical Analysis

The Kruskal–Wallis test was performed for age at diagnosis, the interval between chest CT and [^18^F]FDG PET/CT, the interval between [^18^F]FDG PET/CT and pathological confirmation, GGN size, HU, SUV_max_, and SUV_maxTF_ of the nodule. Fisher’s exact test was performed for the patients’ sex, smoking history, pathological confirmation method, percentage of nodules with SUV_max_ and SUV_maxTF_ >2.5, and the visual positivity rate in each pathological group. The Wilcoxon signed-rank test was performed to determine the significance of changes in SUV_max_ when tissue fraction correction was performed in each pathological group. The McNemar test was performed to ascertain if the number of GGNs with a SUV_max_ of ≥2.5 showed a significant increase when tissue fraction correction was performed. A Mann–Whitney U-test or a Fisher’s exact test was performed for SUV_max_, SUV_maxTF_, and visual positivity based on the epidermal growth factor receptor (EGFR) status. A *p*-value of <0.05 was considered as indicative of statistical significance. Statistical analyses were performed using IBM SPSS Statistics for Windows (Version 27.; IBM Corp. Armonk, NY, USA) and VassarStats (http://www.vassarstats.net (accessed on 8 May 2022)). The post-hoc test was performed with Bonferroni correction.

## 3. Results

### 3.1. Patient Characteristics

A total of 38 patients were enrolled at Hallym University Sacred Heart Hospital (n = 29) and Kangnam Sacred Heart Hospital (n = 9), and a total of 40 GGNs (36 patients with one GGN and 2 patients with two GGNs) were classified according to their pathology. Of the 40 GGNs, 25 were adenocarcinomas, 9 were MIAs, and 6 were AISs. As for the histological subtypes of the 40 GGNs, 28 were lepidic predominant type, 4 were acinar predominant type, 3 were mixed with lepidic and acinar types, 1 was papillary type, and the remaining 4 had an unconfirmed histological subtype. In addition, among the 40 GGNs, 31 and 26 were tested for EGFR and anaplastic lymphoma kinase mutation, respectively, of which 14 (45%) and none were positive, respectively. Among the three pathological groups, there were no significant differences in age, sex, smoking history, nodule size, HU, the interval between chest CT and [^18^F]FDG PET/CT, the interval between [^18^F]FDG PET/CT and pathological confirmation, and the pathological confirmation method (Table 1). Further, none of the patients received treatment, including systemic chemotherapy, which could affect the [^18^F]FDG uptake of GGN before [^18^F]FDG PET/CT.

### 3.2. Chest CT and [^18^F]FDG PET/CT Characteristics

The chest CT and [^18^F]FDG PET/CT characteristics in each pathological group are shown in Table 2. In the visual analysis, the positivity rate was 88% (highest) for adenocarcinoma, 44% for MIA, and 17% (lowest) for AIS. In the post-hoc test, there was a significant difference in positivity rates between adenocarcinoma and AIS (*p* = 0.002). Both before and after tissue fraction correction, the SUV_max_ values were in the order of adenocarcinoma > MIA > AIS, with a significant difference between adenocarcinoma and AIS (*p* = 0.012 and *p* = 0.008, respectively). After tissue fraction correction, the median SUV_max_ was increased by 85% (*p* < 0.001), and the positivity rate of [^18^F]FDG PET/CT, with an SUV_max_ cutoff value of 2.5, also increased significantly from 5% to 50% (*p* < 0.001). No significant difference was observed in SUV_max_, SUV_maxTF_, or visual positivity based on the EGFR mutation status (*p* = 0.827, *p* = 0.891, and *p* = 1.000, respectively). Representative cases are shown in Figure 2 and Figure 3.

## 4. Discussion

This appears to be the first study to attempt evaluating [^18^F]FDG uptake by correcting tissue fraction in malignant pure GGNs. Tissue fraction correction was first introduced by Lambrou et al. to exclude the effect of heterogeneous density in measuring lung [^18^F]FDG uptake in patients with interstitial lung disease [17]. We believed that [^18^F]FDG uptake, excluding the air fraction of GGN, could be measured by applying Lambrou’s method because GGNs contain a high air fraction, and the density varies among GGNs. However, it was unknown which value was appropriate to apply to HU_Tissue_ in the formula, and we assumed that the tissue fraction constituting GGN would have a density similar to that of other solid organs, such as the liver; therefore, we applied a value of 50, as applied by Bondue et al. [19]. As expected, when this method was used, SUV_maxTF_ increased the sensitivity of detecting a malignant pure GGN, and adenocarcinoma expressed as GGNs showed high sensitivity on both visual (88%) and semiquantitative analyses after tissue fraction correction (60%).

The pure GGNs enrolled in this study were confirmed to be adenocarcinoma, MIA, and AIS in pathological analysis. In 2011, Travis et al. reclassified lung adenocarcinoma as follows: (1) if there is a small localized adenocarcinoma of <3 cm characterized by lepidic growth along with the alveolar structure, it is classified as AIS (formerly called bronchioloalveolar carcinoma); (2) if a nodule has papillary, micropapillary, solid growth pattern, or infiltration into the myofibroblastic stroma with an invasion of <5 mm, it is classified as an MIA; and (3) if there is an invasion of >5 mm, invasion of lymphatics, blood vessels, pleura, or presence of tumor necrosis, the nodule is classified as an invasive adenocarcinoma [20]. Therefore, the invasiveness is in the order of adenocarcinoma > MIA > AIS. Similarly, our study also showed [^18^F]FDG positivity, SUV_max_, and SUV_maxTF_ in the order of adenocarcinoma > MIA > AIS. Therefore, we believe that [^18^F]FDG PET/CT reflects the histological invasiveness of GGNs.

In other studies, the false-negative rate of malignant pure GGNs has been reported as high as 90–100% [13,14,15,16], which could be attributed to the high proportion of <1 cm nodules [13,21], the strict criterion of [^18^F]FDG uptake positivity (SUV_max_ ≥ 2.5 [14], or a higher [^18^F]FDG uptake than that of mediastinal blood pool activity [15]). To avoid high false-negative rates due to a small size or a high standard of positive criteria, we evaluated only pure >1 cm GGNs and set the positivity criteria to be [^18^F]FDG uptake higher than background lung activity in visual analysis. As a result, the positivity rate of visual analysis was 68% (88% for adenocarcinoma), whereas the positivity rate in other studies, where an SUV_max_ of 2.5 was set as the cutoff, was very low at 5%. When tissue fraction correction was applied, the sensitivity increased by 50% (60% for adenocarcinoma) despite the high SUV_max_ cutoff of 2.5, which is higher than that reported in previous studies.

SUV_maxTF_ of ≥2.5 showed higher sensitivity (50%) than SUV_max_ of ≥2.5 (5%); however, it was lower than the sensitivity of visual analysis (68%). Therefore, to sensitively predict the malignancy of pure GGN, the results of the visual analysis should be regarded as important. Although the specificity was not available in this study, future studies should be conducted on whether SUV_maxTF_ has better specificity than visual analysis.

Vesselle et al. reported different mean SUV_max_ values according to the histology of lung cancer (large cell, 12.6 ± 5.5; squamous, 11.7 ± 4.5; adenocarcinoma, 9.2 ± 5.8; bronchioloalveolar carcinoma, 3.2 ± 1.7) [22]. In our study, invasive adenocarcinoma showed a relatively low SUV_max_ after the tissue fraction correction (mean SUV_maxTF_ = 3.2 ± 2.4). According to Yoshizawa et al., among the subtypes of invasive adenocarcinoma divided by their growth patterns, solid and micropapillary type adenocarcinomas showed poor prognosis, with a 5-year disease-free survival of 67–76%. On the other hand, acinar, papillary, and lepidic types showed 5-year disease-free survivals ranging 83–90%, with intermediate clinical behavior [23]. These growth patterns are known as stage-independent prognostic indicators [24]; Moon et al. reported that no micropapillary or solid components were found in pure GGNs [25]. In our study, among 36 GGNs with confirmed histological subtypes, the lepidic predominant type was the most common subtype at 70%, followed by acinar predominant, mixed lepidic and acinar, and papillary types. Only two GGNs contained a tiny proportion of micropapillary type (<5% of cancer lesions). No GGNs showed a solid component. Owing to these differences in the histological subtypes, the SUV_max_ may have been lower for GGNs than for solid lung cancer despite tissue fraction correction. In addition, it is well known that the growth rate is lower for GGNs than for solid nodules or mixed GGNs. According to Hasegawa et al., the median volume doubling time of pure GGNs is about 831 days, which is much longer than that for mixed GGNs (about 457 days), suggesting that pure GGNs are relatively indolent [26]. Slow-growing tumors are thought to have a low metabolic demand because of a low number of metabolically active malignant cells [11,27], which may be one reason why the SUV_max_ was low even after tissue fraction correction.

McDermott et al. reported that the mean SUV_max_ of 21 malignant GGNs was 0.8 ± 0.3, which is lower than that of 106 benign GGNs (1.6 ± 1.5, *p* = 0.002) [28]; however, malignant GGNs showed a mean SUV_max_ of 1.5 ± 1.2 in our study, which is significantly higher than that reported in their study (*p* = 0.011). We performed a biopsy on all the 40 malignant GGNs, whereas McDermott et al. performed biopsy confirmation on only 3 out of 127 GGNs. Consequently, it is difficult to clarify the discrepancy between the two studies. Had McDermott et al. included many malignant GGNs with low [^18^F]FDG uptakes, such as AIS and MIA, it would have been possible to show such low [^18^F]FDG uptake of malignant GGNs.

In general, CT attenuation, presenting as GGNs, is higher for invasive adenocarcinoma than for the precursor [29,30]. Recent studies have reported that the SUV_max_ positively correlates with the size, cellularity, and aggressiveness of the lesion but negatively correlates with the percentage of ground-glass opacity [20,23,27,31,32,33]. In our study, there was no significant difference in HUs between the three pathological groups, but significant differences were found in the SUV_max_ and SUV_maxTF_. Thus, [^18^F]FDG PET/CT may be more beneficial in analyzing GGNs than HU.

This study had several limitations. First, there was no benign lesion among the GGNs included in this study; thus, specificity could not be calculated. Because of the slow-growing nature of GGNs, it was difficult to determine the malignancy of a nodule using follow-up imaging as at least a 5-year follow-up is required for subsolid nodules as per the Fleischner Society 2017 Guidelines [10]. Second, respiratory gating was not performed. If misregistration occurred, visual and semiquantitative analyses were performed, assuming that the visually discernible [^18^F]FDG uptake near the GGN was the [^18^F]FDG uptake of the GGN. However, [^18^F]FDG uptake could have been underestimated due to inaccurate attenuation correction.

## 5. Conclusions

Tissue fraction correction and visual analysis increased the sensitivity of predicting the malignancy of pure GGNs larger than 1 cm on [^18^F]FDG PET/CT.

## Figures and Tables

**Figure 1 diagnostics-12-01292-f001:**
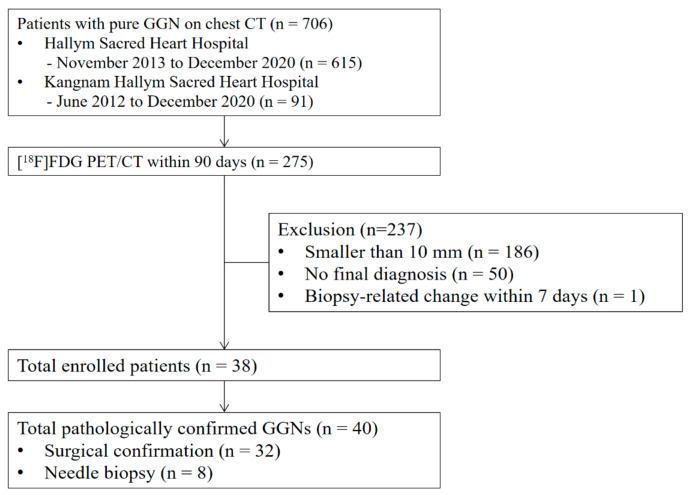
Flow diagram of patient enrollment.

**Figure 2 diagnostics-12-01292-f002:**
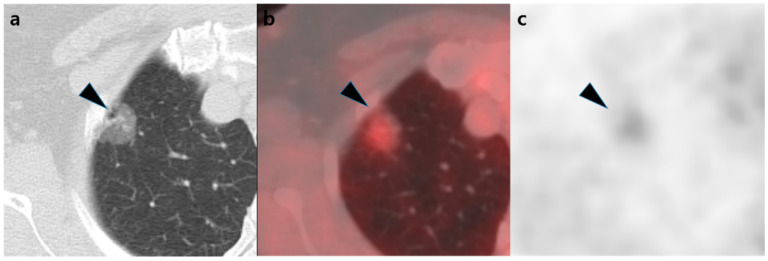
An example of an adenocarcinoma. A 58-year-old woman’s chest computed tomography showing a 20 mm ground-glass nodule (black arrow head) (**a**) with visually positive [^18^F]FDG uptake (**b**,**c**). The Hounsfield units value of the nodule was −436, and the SUV_max_ increased from 1.99 to 3.70 after tissue fraction correction, which was higher than the cutoff value of 2.50. The nodule was diagnosed as adenocarcinoma after surgery.

**Figure 3 diagnostics-12-01292-f003:**
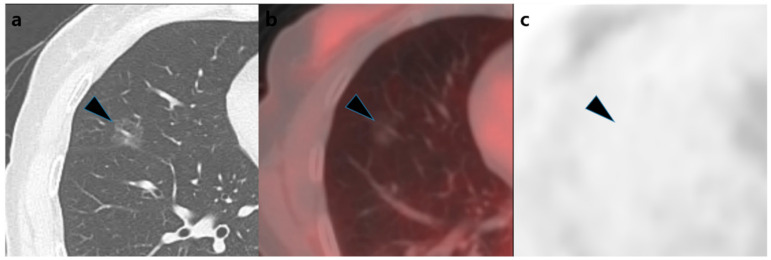
An example of adenocarcinoma in situ. A 60-year-old woman’s chest computed tomography showing a 17 mm ground-glass nodule (black arrow head) (**a**) with visually negative [^18^F]FDG uptake (**b**,**c**). The Hounsfield units value of the nodule was −644, and the SUV_max_ increased from 0.52 to 1.53 after tissue fraction correction, which was lower than the cutoff value of 2.50. The nodule was diagnosed as adenocarcinoma in situ after surgery.

**Table 1 diagnostics-12-01292-t001:** Patients’ characteristics.

Characteristics	n = 38
Age at diagnosis, year, median (Q1–Q3)	64.0 (59.0–68.0)
Sex	
Male, n (%)	13 (34%)
Female, n (%)	25 (66%)
Smoking history	
Current smoker, n (%)	3 (8%)
Former smoker, n (%)	0 (0%)
Nonsmoker, n (%)	35 (92%)
Reason for [^18^F]FDG PET/CT	
For ground-glass nodule evaluation, n (%)	35 (92%)
For other malignancy evaluation, n (%)	3 (8%)
Interval between CT and [^18^F]FDG PET/CT, days, median (Q1–Q3)	17.5 (11.5–24.8)
Interval between [^18^F]FDG PET/CT and biopsy, days, median (Q1–Q3)	5.0 (2.0–10.5)

Q1, 25th percentile; Q3, 75th percentile.

**Table 2 diagnostics-12-01292-t002:** Characteristics of the chest CT and [^18^F]FDG PET/CT findings.

Characteristics	Adenocarcinoma (n = 25)	Minimally Invasive Adenocarcinoma (n = 9)	Adenocarcinoma In Situ (n = 6)	Total (n = 40)	*p*
Size of nodule, mm	19.0 (15.0–23.0)	13.3 (10.0–20.0)	14.7 (12.0–16.8)	16.8 (12.0–23.0)	0.125
Hounsfield unit	−437 (−529–−377)	−411 (−631–−313)	−577 (−631–−435)	−437 (−598–−379)	0.406
Method for pathological confirmation					0.227
Needle biopsy	7 (28%)	0 (0%)	1 (17%)	8 (20%)	
Surgery	18 (72%)	9 (100%)	5 (83%)	32 (80%)	
Visual analysis of [^18^F]FDG PET/CT					0.001 *
Positive	22 (88%)	4 (44%)	1 (17%)	27 (68%)	
Semi-quantitative analysis of [^18^F]FDG PET/CT					
SUV_max_	1.3 (1.1–1.8)	1.1 (0.7–1.8)	0.6 (0.5–0.9)	1.3 (0.8–1.8)	0.033 *
SUV_maxTF_	2.6 (2.2–3.1)	2.2 (1.9–2.9)	1.6 (1.5–1.7)	2.4 (1.9–3.0)	0.018 *
SUV_max_ ≥ 2.5	2 (8%)	0 (0%)	0 (0%)	2 (5%)	0.990
SUV_maxTF_ ≥ 2.5	15 (60%)	4 (44%)	1 (17%)	20 (50%)	0.195

Data shown as median (25th percentile–75th percentile) or n (%). SUV_max_, maximum standardized uptake value; SUV_maxTF_, tissue-fraction–corrected SUV_max_. * *p* < 0.05 was considered statistically significant.

## Data Availability

Not applicable.

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
