# Peer review of "Tissue Fraction Correction and Visual Analysis Increase Diagnostic Sensitivity in Predicting Malignancy of Ground-Glass Nodules on [18F]FDG PET/CT: A Bicenter Retrospective Study"

_diagnostics, 2022, doi:10.3390/diagnostics12051292_

Round 1
Reviewer 1 Report
The Authors describe the role of tissue fraction correction and visual analysis in predicting malignancy of ground-glass lung nodules on [18F]FDG PET/CT.
It is a bi-center retrospective study
The idea is interesting
Some points need to be clarified and/or modified:
- The study population includes 40 pathologically confirmed malignant ground-glass nodules in 38 patients. No benign lesion was included. This is an important limitation because specificity could not be calculated. The period of enrollment is long (2013-2020) and the initial population is large (706 pts): it is possible and appropriate to find a group of patients with benign lesions confirmed by adequate follow up and include it in the study.
- The table 2 is too detailed: some details are not fundamental for the paper aim (i.e. location, histological subtype, decimals in the percentages)
- According to the results reported on Table 2, “visual analysis” is better than “semiquantitative analysis”: what is the diagnostic add value of SUVmaxTF and SUVmaxTF ≥ 2.5? This point has to be explained and discussed
Author Response
We thank the reviewers for their helpful comments on our manuscript. We have made every effort to address their concerns and have revised the manuscript according to their suggestions. Furthermore, we performed additional analyses and reviewed more papers to supplement the results reported in our manuscript as per the reviewers’ recommendations. The revisions made and our responses to the reviewers’ comments are presented in blue text in the attached file.
Reviewer #1: The Authors describe the role of tissue fraction correction and visual analysis in predicting malignancy of ground-glass lung nodules on [18F]FDG PET/CT.
It is a bi-center retrospective study
The idea is interesting
Some points need to be clarified and/or modified:
The study population includes 40 pathologically confirmed malignant ground-glass nodules in 38 patients. No benign lesion was included. This is an important limitation because specificity could not be calculated. The period of enrollment is long (2013-2020) and the initial population is large (706 pts): it is possible and appropriate to find a group of patients with benign lesions confirmed by adequate follow-up and include it in the study.
→ The initial study population included 706 patients. However, benign GGN found by imaging follow-up was not initially subjected to [18F]FDG PET/CT. [18F]FDG PET/CT was generally performed when surgery was already planned owing to the high suspicion of malignancy by clinicians. Considering that the enrolled patients had a GGN of at least 1 cm confirmed at least twice on CT, it is understandable why there was no benign GGN with [18F]FDG PET/CT. Furthermore, unlike solid nodules, GGNs require imaging follow-up for at least 5 years based on the Fleischner Society 2017 Guidelines. Because this study was conducted on patients for 7 years, most were followed up for <5 years or lost during follow-up. Our authors thoroughly searched the electronic medical records but could not find benign GGN with [18F]FDG PET/CT. Therefore, we had to leave it as a limitation. We modified the following sentences for clarification:
- Page 2 line 75: No patients with benign or malignant tumor could be determined by imaging follow-up for >5 years.
- Page 7 line 276: Because of the slow-growing nature of GGNs, it was difficult to determine the malignancy of a nodule using follow-up imaging as at least a 5-year follow-up is required for subsolid nodules as per the Fleischner Society 2017 Guidelines [10].
The table 2 is too detailed: some details are not fundamental for the paper aim (i.e. location, histological subtype, decimals in the percentages)
→ The table has been modified as per the reviewer’s advice. Values after decimal points were also removed. The contents deleted from the table were replaced with simple sentences as follows:
- Page 4 line 146: As for the histological subtypes of the 40 GGNs, 28 were lepidic predominant type, 4 were acinar predominant type, 3 were mixed with lepidic and acinar types, 1 was papillary type, and the remaining 4 had unconfirmed histological subtype.
According to the results reported on Table 2, “visual analysis” is better than “semiquantitative analysis”: what is the diagnostic add value of SUVmaxTF and SUVmaxTF ≥ 2.5? This point has to be explained and discussed
→ Thank you for raising a valid point. We have accordingly added the following sentences in the Discussion section:
- Page 7 line 231: SUVmaxTF of ≥5 showed higher sensitivity (50%) than SUVmax of ≥2.5 (5%); however, it was lower than the sensitivity of visual analysis (68%). Therefore, to sensitively predict the malignancy of pure GGN, the results of the visual analysis should be regarded as important. Although the specificity was not available in this study, future studies should be conducted on whether SUVmaxTF has better specificity than visual analysis.
Reviewer 2 Report
The authors investigated investigated the role of [18F]FDG positron-emission tomography/computed tomography (PET/CT) in evaluating ground-glass nodules (GGNs) by visual analysis and tissue fraction correction. When using a cut-off value of 2.5, the positivity rate of GGNs was significantly higher in SUVmaxTF than in SUVmax (50.0% vs. 5.0%, p < 0.001). According to the authors, the diagnostic sensitivity of [18F]FDG PET/CT in predicting malignancy of lung GGN was improved by tissue fraction correction and visual analysis.
Major issues
As stated in the limitations, missing follow-up is a substantial lack in this study. Is there no follow-up at all? Maybe a clinical follow-up, if not imaging? This should be checked and further reported.
All GGNs were pathologically confirmed either surgically or by needle biopsy (figure 1). Were gene mutations in the GGNs analyzed (such as EGFR, ERBB2, and KRAS)? This should be stated and the results presented.
Minor issues
If a mutation analysis was performed, the results should be correlated with the FDG PET/CT results. Are there differences in the PET/CT presentation of NSCLC with different mutations? This should be shown and discussed.
Did any patient receive treatment before FDG PET/CT was performed? This should be stated.
The conclusion “Tissue fraction correction and visual analysis increased the sensitivity of predicting the malignancy of pure GGNs on [18F]FDG PET/CT.” is difficult to state when nodules < 1cm were excluded in this study. The conclusion should be changed.
The entire manuscript should be checked for spell and punctuation issues (e.g. line 68 “The”, line 69 “wreting”).

Author Response
We thank the reviewers for their helpful comments on our manuscript. We have made every effort to address their concerns and have revised the manuscript according to their suggestions. Furthermore, we performed additional analyses and reviewed more papers to supplement the results reported in our manuscript as per the reviewers’ recommendations. The revisions made and our responses to the reviewers’ comments are presented in blue text in the attached file.
Reviewer #2: The authors investigated the role of [18F]FDG positron-emission tomography/computed tomography (PET/CT) in evaluating ground-glass nodules (GGNs) by visual analysis and tissue fraction correction. When using a cut-off value of 2.5, the positivity rate of GGNs was significantly higher in SUVmaxTF than in SUVmax (50.0% vs. 5.0%, p < 0.001). According to the authors, the diagnostic sensitivity of [18F]FDG PET/CT in predicting malignancy of lung GGN was improved by tissue fraction correction and visual analysis.
Major issues
As stated in the limitations, missing follow-up is a substantial lack in this study. Is there no follow-up at all? Maybe a clinical follow-up, if not imaging? This should be checked and further reported.
→ Unlike solid nodules, GGN requires imaging follow-up for at least 5 years based on the Fleischner Society 2017 Guidelines. Because this study was conducted on patients for 7 years, most were followed up for <5 years or lost during follow-up. Therefore, we had no choice but to classify them as “no final diagnosis” in the flow diagram of patient enrollment (Figure 1). We modified the following sentences for clarification.
- Page 2 line 75: No patients with benign or malignant tumor could be determined by imaging follow-up for >5 years.
- Page 7 line 276: Because of the slow-growing nature of GGNs, it was difficult to determine the malignancy of a nodule using follow-up imaging as at least a 5-year follow-up is required for subsolid nodules as per the Fleischner Society 2017 Guidelines [10].
All GGNs were pathologically confirmed either surgically or by needle biopsy (figure 1). Were gene mutations in the GGNs analyzed (such as EGFR, ERBB2, and KRAS)? This should be stated and the results presented.
→ As suggested by the reviewer, we analyzed the gene mutation status. Among 40 GGNs, 31 and 26 were tested for EGFR and ALK mutation, respectively. No significant difference was observed in EGFR mutation status between the adenocarcinoma, MIA, and AIS groups (p = 0.292, see the table below). No patient had ALK mutation in our study.
|
Adenocarcinoma (n = 25) |
MIA (n = 9) |
AIS (n = 6) |
Total (n = 40) |
EGFR mutation |
15 (60%) |
2 (22%) |
0 (0%) |
17 (43%) |
No EGFR mutation |
9 (36%) |
3 (33%) |
2 (33%) |
14 (35%) |
Unknown |
1 (4%) |
4 (44%) |
4 (67%) |
9 (23%) |
|
|
|
|
|
We simplified the table according to Reviewer#1’s suggestion. Therefore, we added the following sentence instead of modifying Table 2.
- Page 4 line 149: In addition, among the 40 GGNs, 31 and 26 were tested for EGFR and anaplastic lymphoma kinase mutation, respectively, of which 14 (45%) and none were positive, respectively.
Minor issues
If a mutation analysis was performed, the results should be correlated with the FDG PET/CT results. Are there differences in the PET/CT presentation of NSCLC with different mutations? This should be shown and discussed.
→ We further evaluated the differences of SUVmax, SUVmaxTF, and visual positivity as per the EGFR mutation status as you suggested and added the following sentence:
- Page 4 line 135: Mann–Whitney U-test or Fisher’s exact was performed for SUVmax, SUVmaxTF, and visual positivity based on the EGFR status.
- Page 4 line 170: No significant difference was observed in SUVmax, SUVmaxTF, and visual positivity based on EGFR mutation status (p = 0.827, p = 0.891, and p = 1.000, respectively).
Did any patient receive treatment before [18F]FDG PET/CT was performed? This should be stated.
→ Thirty seven patients did not receive any treatment before 30 months of [18F]FDG PET/CT imaging. The remaining one patient had a bilateral partial mastectomy for bilateral breast cancer, and [18F]FDG PET/CT was performed while receiving radiation therapy on both breasts. However, the GGN was located outside the radiation therapy field. The GGN was negative on both visual and semi-quantitative analyses. The GGN was surgically confirmed as AIS. Systemic treatment, such as chemotherapy, was not performed. We added the following sentence in the manuscript:
- Page 4 line 154: Further, none of the patients received treatment, including systemic chemotherapy, which could affect the [18F]FDG uptake of GGN before [18F]FDG PET/CT.
The conclusion “Tissue fraction correction and visual analysis increased the sensitivity of predicting the malignancy of pure GGNs on [18F]FDG PET/CT.” is difficult to state when nodules < 1cm were excluded in this study. The conclusion should be changed.
→ Thanks for the valuable comment. We changed the conclusion as you suggested.
- Page 8 line 285: Tissue fraction correction and visual analysis increased the sensitivity of predicting the malignancy of pure GGNs larger than 1cm on [18F]FDG PET/CT.
The entire manuscript should be checked for spell and punctuation issues (e.g. line 68 “The”, line 69 “wreting”).
→ Thank you for bringing this point to our attention. We have revised our manuscript for language and grammar and enhanced its clarity based on suggestions from a native English editing service.
Reviewer 3 Report
The investigation of [18F]FDG PET/CT in predicting malignancy of lung GGN was improved by tissue fraction correction and visual analysis is an interesting idea and the result are well written.
Author Response
We are grateful for your thoughtful review.
Reviewer 4 Report
the paper is on using two methods including tissue fraction correction and visual analysis for predicting the malignancy of ground-glass 3
nodules in [18F]FDG PET/CT. The authors explained their experiments and results in a professional way and I think this paper can be helpful for researchers in PET/CT field after several revisions:
- Rewrite the title, it is difficult to follow
- the introduction is too short. please add a couple of paragraphs explaining your methods and the novelty of your research
- please rewrite this sentence "With the development of thin section and high-resolution chest computed 36
tomography (CT) [1–3], the detection rate of ground-glass nodules (GGN) has also been 37
increasing." - please clarify the betterness of your study compared to previous studies in this field in the discussion section?
- this sentence does not sound scientific; please rewrite it: Therefore, we think that [18F]FDG PET/CT reflects the histological invasiveness of GGN.
- Is the visual analysis too time-consuming? what is your suggestion for that? can artificial intelligence be helpful in this regard? discuss this in the discussion.
- The conclusion is too short. Please add more information
Author Response
We would like to thank the reviewer for detailed comments and hard work. However, on May 6th, we had already been instructed to revise within 10 days based on comments already made by 3 other reviewers. When we received this review, we had already entrusted with the English proofreading. We will respond when the next opportunity arises according to the decision of the editor. Thank you very much for your review.
Round 2
Reviewer 1 Report
The Authors responded adequately to the reviewers' comments.
Reviewer 2 Report
The authors included all required changes in the revised version.